# Learning to Rank Learning Curves

## Abstract

Many automated machine learning methods, such as those for hyperparameter and neural architecture optimization, are computationally expensive because they involve training many different model configurations. In this work, we present a new method that saves computational budget by terminating poor configurations early on in the training. In contrast to existing methods, we consider this task as a ranking and transfer learning problem. We qualitatively show that by optimizing a pairwise ranking loss and leveraging learning curves from other data sets, our model is able to effectively rank learning curves without having to observe many or very long learning curves. We further demonstrate that our method can be used to accelerate a neural architecture search by a factor of up to 100 without a significant performance degradation of the discovered architecture. In further experiments we analyze the quality of ranking, the influence of different model components as well as the predictive behavior of the model.

## 1 Introduction

A method commonly used by human experts to speed up the optimization of neural architectures or hyperparameters is the early termination of iterative training processes that are unlikely to improve the current solution. A common technique to determine the likelihood of no improvement is to compare the learning curve of a new configuration to the one of the currently best configuration. This idea can also be used to speed up automated machine learning processes. For this purpose, it is common practice to extrapolate the partial learning curve in order to predict the final performance of the currently investigated model. Current extrapolation techniques have several weaknesses that make them unable to realize their full potential in practice. Many of the methods require sufficient sample learning curves to make reliable predictions (Chandrashekaran & Lane, 2017; Klein et al., 2017; Baker et al., 2018). Thus, the extrapolation method for the first candidates can not be used yet, which means more computational effort. Other methods do not have this disadvantage, but require sufficiently long learning curves to make reliable predictions which again means unnecessary overhead (Domhan et al., 2015). Many of these methods also do not take into account other information such as the hyperparameters of the model being examined or its network architecture.

We address the need for sample learning curves by devising a transfer learning technique that uses learning curves from other problems. Since the range of accuracy varies from data set to data set, we are forced to consider this in our modeling. But since we are not interested in predicting the performance of a model anyway, we use a ranking model that models the probability that the model currently being investigated surpasses the best solution so far. This does not only solve the problem but also provides a better modeling of the actual task. In order to be able to make reliable predictions for short learning curves, we consider further characteristics of the model such as its network architecture. We compare our ranking method with respect to a ranking measure against different methods on five different image classification data sets. We also show that our method is capable of significantly accelerating a neural architecture search. Furthermore, we conduct several ablation studies to provide a better motivation of our model and its behavior.

## 2 Related work

Most of the prior work for learning curve prediction is based on the idea of extrapolating the partial learning curve by using a combination of continuously increasing basic functions.

Domhan et al. (2015) define a set of 11 parametric basic functions, estimate their parameters and combine them in an ensemble. Klein et al. (2017) propose a heteroscedastic Bayesian model which learns a weighted average of the basic functions. Chandrashekaran & Lane (2017) do not use basic functions but use previously observed learning curves of the current data set. An affine transformation for each previously seen learning curve is estimated by minimizing the mean squared error with respect to the partial learning curve. The best fitting extrapolations are averaged as the final prediction. Baker et al. (2018) use a different procedure. They use support vector machines as sequential regressive models to predict the final accuracy based on features extracted from the learning curves, its gradients, and the neural architecture itself.

The predictor by Domhan et al. (2015) is able to forecast without seeing any learning curve before but requires observing more epochs for accurate predictions. The model by Chandrashekaran & Lane (2017) requires seeing few learning curves to extrapolate future learning curves. However, accurate forecasts are already possible after few epochs. Algorithms proposed by Klein et al. (2017); Baker et al. (2018) need to observe many full-length learning curves before providing any useful forecasts. However, this is prohibiting in the scenarios where learning is time-consuming such as in large convolutional neural networks.

All previous methods for automatically terminating iterative learning processes are based on methods that predict the learning curve. Ultimately, however, we are less interested in the exact learning curve but rather whether the current learning curve leads to a better result. This way of obtaining a ranking is referred to as pointwise ranking methods (Liu, 2011). They have proven to be less efficient than pairwise ranking methods which directly optimize for the objective function (Burges et al., 2005). Yet, we are the first to consider a pairwise ranking loss for this application.

There are some bandit-based methods which leverage early termination as well. Successive Halving (Jamieson & Talwalkar, 2016) is a method that trains multiple models with settings chosen at random simultaneously and terminates the worst performing half in predefined intervals. Hyperband (Li et al., 2017) identifies that the choice of the intervals is vital and therefore proposes to run Successive Halving with different intervals. BOHB (Falkner et al., 2018) is an extension of Hyperband which proposes to replace the random selection of settings with Bayesian optimization.

The use of methods that terminate less promising iterative training processes early are of particular interest in the field of automated machine learning, as they can, for example, significantly accelerate hyperparameter optimization. The most computationally intensive subproblem of automated machine learning is Neural Architecture Search (Zoph & Le, 2017), the optimization of the neural network topology. Our method is not limited to this problem, but as it is currently one of the major challenges in automated machine learning, we use this problem as a sample application in our evaluation. For this particular application, optimization methods that leverage parameter sharing (Pham et al., 2018) have become established as a standard method. Here, a search space spanning, overparametrised neural network is learned and finally used to determine the best architecture. DARTS (Liu et al., 2019) is one of the last extensions of this idea which uses a continuous relaxation of the search space, which allows learning of the shared parameters as well as of the architecture by means of gradient descent. This method can be a strong alternative to early termination methods but does not transfer to general machine learning or arbitrary architecture search spaces.

## 3   LEARNING CURVE RANKING

With *learning curve* we refer to the function of qualitative performance with growing number of iterations of an iterative learning algorithm. We use the term *final learning curve* to explicitly denote the entire learning curve, $y_1, \ldots, y_L$, reflecting the training process from beginning to end. Here, $y_i$ is a measure of the performance of the model (e.g., classification accuracy), which is determined at regular intervals. Contrary, a *partial learning curve*, $y_1, \ldots, y_l$, refers to learning curves that are observed only up to a time $l$. We visualize the concepts of the terms in Figure 1.

There is a set of automated early termination methods which follow broadly the same principle. The first model is trained to completion and is considered as the current best model $m^{\max}$ with its performance being $y^{\max}$. For each further model $m_i$, the probability that it is better than the current best model, $p\left(m_i > m^{\max}\right)$, is monitored at periodic intervals during training. If it is below a given threshold, the model's training is terminated (see Algorithm 1). All existing methods rely on a two-

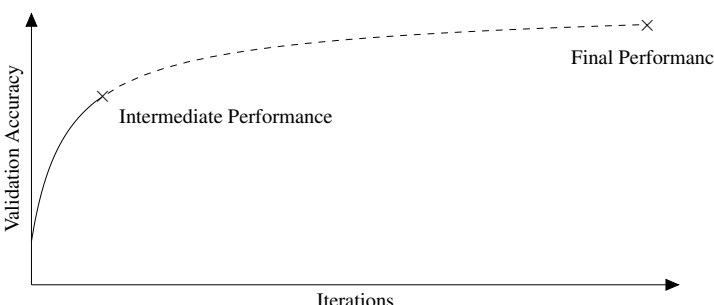

Figure 1: Learning curve prediction tries to predict from the partial learning curve (solid line) the final performance.

step approach to determine this probability which involves extrapolating the partial learning curve and a heuristic measure. Instead of the two-step process to determine the probability, we propose LCRankNet to predict the probability that a model $m_i$ is better than $m_j$ directly. LCRankNet is based on a neural network $f$ which considers model characteristics and the partial learning curve $\mathbf{x}_i$ as input. We define the probability that $m_i$ is better than $m_j$ as

$$p(m_i > m_j) = \hat{p}_{i,j} = \frac{e^{f(\mathbf{x}_i)-f(\mathbf{x}_j)}}{1 + e^{f(\mathbf{x}_i)-f(\mathbf{x}_j)}}. \tag{1}$$

Using the logistic function is an established modelling approach in the learning-to-rank community (Burges et al., 2005). Given a set of final learning curves for some models, the estimation of the posterior values $p_{i,j}$ between these models is trivial since we know whether $m_i$ is better than $m_j$ or not. Therefore, we set

$$p_{i,j} = \begin{cases} 1 & \text{if } m_i > m_j \\ 0.5 & \text{if } m_i = m_j \\ 0 & \text{if } m_i < m_j . \end{cases} \tag{2}$$

We minimize the cross-entropy loss

$$\sum_{i,j} -p_{i,j} \log \hat{p}_{i,j} - (1 - p_{i,j}) \log(1 - \hat{p}_{i,j}) \tag{3}$$

to determine the parameters of $f$. Now the probability $p(m_i > m_j)$ can be predicted for arbitrary pairs $m_i$ and $m_j$ using Equation (1).

---

**Algorithm 1** Early Termination Method

---

**Input:** Data set $d$, model $m$, performance of best model $m^{\max}$ so far $y^{\max}$.
**Output:** Learning curve.
  1: **for** $l \leftarrow 1 \ldots L$ **do**
  2:     Train $m$ on $d$ for a step and observe a further part of the learning curve $y_t$.
  3:     **if** $\max_{1 \leq i \leq l} y_i > y^{\max}$ **then**
  4:         **continue**
  5:     **else if** $p(m > m^{\max}) \leq \delta$ **then**
  6:         **return y**
  7: **return y**

---

### 3.1 RANKING MODEL TO LEARN ACROSS DATA SETS

We have defined the prediction of our model in Equation (1). It depends on the outcome of the function $f$ which takes a model representation $\mathbf{x}$ as input. The information contained in this representation depends on the task at hand. Since we will consider Neural Architecture Search, the representation consists of three different parts. First, the partial learning curve $y_1, \ldots, y_l$. Second, the description of the model's architecture which is a sequence of strings representing the layers it is

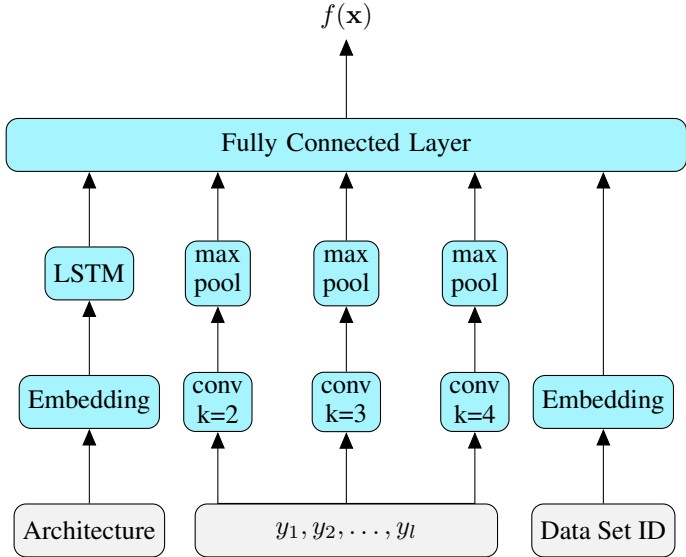

Figure 2: LCRankNet has three different components, each dealing with one type of input: partial learning curve, architecture encoding and data set ID.

comprised of. Third, a data set ID to indicate on which data set the corresponding architecture and learning curve was trained and observed.

We model $f$ using a neural network and use special layers to process the different parts of the representation. The learned representation for the different parts are finally concatenated and fed to a fully connected layer. The architecture is visualized in Figure 2. We will now describe the different components in detail.

**Learning curve component**    A convolutional neural network is used to process the partial learning curve. We consider up to four different convolutional layers with different kernel sizes. The exact number depends on the length of the learning curve since we consider no kernel sizes larger than the learning curve length. Each convolution is followed by a global max pooling layer and their outputs are concatenated. This results in the learned representation for the learning curve.

**Architecture component**    For our learning curve prediction we use architectures of the popular NASNet search space (Zoph et al., 2018). We use the most common configuration which consists of two cells and every cell of five blocks. This results into 40 choices to be made which is encoded as a sequence of integers. Please refer to Zoph et al. (2018) for further details regarding the search space. We learn an embedding for every choice. An LSTM takes this embedding and generates the architecture embedding.

**Data set component**    As we intend to learn from learning curves of other data sets, we include data set embeddings as one of the components. Every data set has its own embedding and it will be used whenever a learning curve observed on this data set is selected. If the data set is new, then the corresponding embedding is initialized at random. As the model observes more learning curves from this data set, it improves the data set embedding. The embedding helps us to model data-set-specific peculiarities in order to adjust the ranking if necessary.

**Technical details**    During the development process, we found that the architecture component leads to instabilities during training. To avoid this, we regularize the output of the LSTM layer by using it as an input to an autoregressive model, similar to a sequence-to-sequence model (Sutskever et al., 2014) that recreates the original description of the architecture. In addition, we use the attention mechanism (Bahdanau et al., 2015) to facilitate this process. All parameters of the layers in $f$ are trained jointly by means of Adam (Kingma & Ba, 2015) by minimizing a weighted linear combination of the ranking loss (Equation (3)) and the reconstruction loss with respect to its parameters.

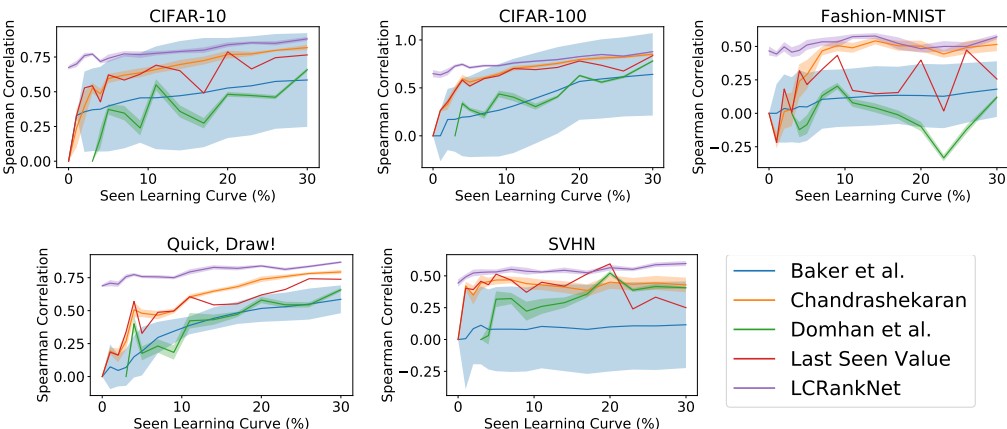

Figure 3: The x-axis indicates the length observed of the learning curve. We report the mean of ten repetitions. The shaded area is the standard deviation. Our method LCRankNet outperforms its competitors on all data sets.

The hyperparameter $\delta$ allows for trading precision vs. recall or search time vs. regret, respectively. There are two extreme case: if we set $\delta \geq 1$ every run is terminated immediately. If we set $\delta \leq 0$ we never terminate any run. For our experiment we set $\delta = 0.45$ which means that if the predicted probability that the new model is better than the best one is below 45%, the run is terminated early.

# 4 EXPERIMENTS

In this section, we first discuss how to create the meta-knowledge and then analyze our model in terms of learning curve ranking and the ability to use it as a way to accelerate Neural Architecture Search. Finally, we examine its individual components and behavior in certain scenarios.

## 4.1 META-KNOWLEDGE

We compare our method to similar methods on five different data sets: CIFAR-10, CIFAR-100, Fashion-MNIST, Quickdraw, and SVHN. We use the original train/test splits if available. Quickdraw has a total of 50 million data points and 345 classes. To reduce the training time, we select a subset of this data set. We use 100 different randomly selected classes and choose 300 examples per class for the training split and 100 per class for the test split. 5,000 random data points of the training data set serve as validation split for all data sets.

To create the meta-knowledge, we choose 200 architectures per data set at random from the NASNet search space (Zoph et al., 2018) such that we train a total of 1,000 architectures. We would like to point out that these are 1,000 unique architectures, there is no architecture that has been trained on several different data sets. Each architecture is trained for 100 epochs with stochastic gradient descent and cosine learning rate schedule without restart (Loshchilov & Hutter, 2017).

We use standard image preprocessing and augmentation: for every image, we first subtract the channel mean and then divide by the channel standard deviation. Images are padded by a margin of four pixels and randomly cropped back to the original dimension. For all data sets but SVHN we apply random horizontal flipping. Additionally, we use Cutout (DeVries & Taylor, 2017).

The following experiments are conducted in a leave-one-data-set-out cross-validation. That means when considering one data set, all meta-knowledge but the one for this particular data set is used as meta-knowledge.

Table 1: Results obtained by the different methods on five different data sets. For both metrics the smaller, the better. Regret reported in percent, time in GPU hours.

| Method | CIFAR-10 | | CIFAR-100 | | Fashion | |
|---|---|---|---|---|---|---|
| | Regret | Time | Regret | Time | Regret | Time |
| No Early Termination | 0.00 | 1023 | 0.00 | 1021 | 0.00 | 1218 |
| Domhan et al. (2015) | 0.56 | 346 | 0.82 | 326 | 0.00 | 460 |
| Li et al. (2017) | 0.22 | 106 | 0.78 | 102 | 0.32 | 132 |
| Baker et al. (2018) | 0.00 | 89 | 0.00 | 77 | 0.00 | 129 |
| Jamieson & Talwalkar (2016) | 0.62 | 62 | 0.00 | 54 | 0.18 | 70 |
| Chandrashekaran & Lane (2017) | 0.62 | 30 | 0.00 | 35 | 0.28 | 41 |
| LCRankNet | 0.22 | 20 | 0.00 | 11 | 0.10 | 19 |

| Method | Quickdraw | | SVHN | |
|---|---|---|---|---|
| | Regret | Time | Regret | Time |
| No Early Termination | 0.00 | 1045 | 0.00 | 1485 |
| Domhan et al. (2015) | 0.44 | 331 | 0.28 | 471 |
| Li et al. (2017) | 0.54 | 109 | 0.00 | 156 |
| Baker et al. (2018) | 0.00 | 107 | 0.00 | 241 |
| Jamieson & Talwalkar (2016) | 0.40 | 60 | 0.28 | 88 |
| Chandrashekaran & Lane (2017) | 0.30 | 82 | 0.06 | 164 |
| LCRankNet | 0.00 | 28 | 0.10 | 74 |

## 4.2 RANKING PERFORMANCE

First, we analyze the quality of the learning curve rankings by different learning curve prediction methods. In this experiment we choose 50 different learning curves at random as a test set. Five random learning curves are used as a training set for every repetition. Each learning curve prediction method ranks the 50 architectures by observing the partial learning curve whose length varies from 0 to 30 percent of the final learning curve. We repeat the following experiment ten times and report the mean and standard deviation of the correlation between the true and the predicted ranking in Figure 3. As a correlation metric, we use Spearman's rank correlation coefficient. Thus, the correlation is 1 for a perfect ranking and 0 for an uncorrelated, random ranking. Our method LCRankNet shows for all data sets better performance. If there are no or only very short partial learning curves available, our method shows the biggest difference to the existing methods. The reason for this is a combination of the consideration of the network architecture together with additional meta-knowledge. We analyze the impact of each component in detail in Section 4.5.

The method of Chandrashekaran & Lane (2017) consistently shows the second best results and in some cases can catch up to the results of our method. The method of Baker et al. (2018) stands out due to the high standard deviation. It is by far the method with the smallest squared error on test. However, the smallest changes in the prediction lead to a significantly different ranking, which explains the high variance in their results. The method of Domhan et al. (2015) requires a minimum length of the learning curve to make predictions. Accordingly, we observe rank correlation values starting from a learning curve length of 4%. Using the last seen value to determine the ranking of learning curves is a simple yet efficient method (Klein et al., 2017). In fact, it is able to outperform some of the more elaborate methods.

## 4.3 ACCELERATING RANDOM NEURAL ARCHITECTURE SEARCH

In this experiment, we demonstrate the utility of learning curve predictors in the search for network architectures. For the sake of simplicity we accelerate a random search in the NASNet search space (Zoph et al., 2018).

The random search samples 200 models and trains each of them for 100 epochs to obtain the final accuracy. In the end, the best of all these models is returned. Now each learning curve predictor

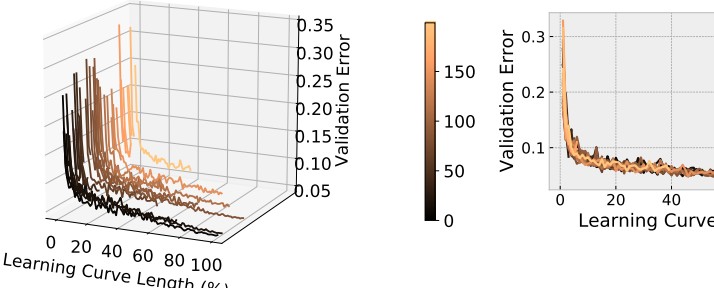

Figure 4: LCRankNet is speeding up architecture search for SVHN. Difference between learning curves is small, making this the hardest task.

iterates over these sampled architectures in the same order and determines at every third epoch if the training should be aborted. For Successive Halving (Jamieson & Talwalkar, 2016) and Hyperband (Li et al., 2017) we follow the algorithm defined by the authors and use the recommended settings. The current best model discovered after iterating over all 200 architectures is returned as the best model for this learning curve predictor. The goal of the methods is to minimize the regret compared to a random search without early stopping and at the same time reduce the computational effort.

One of our observations is that Domhan et al. (2015)'s method underestimates performance when the learning curve is short. As a result, the method ends each training process early after only a few epochs. Therefore, we follow the recommendation of Domhan et al. (2015) and do not end processes before we have seen the first 30 epochs. We summarize the results in Table 1. Our first observation is that all methods accelerate the search with little regret. Here we define regret as the difference between the accuracy of the model found by the random search and the accuracy of the model found by one of the methods. Not surprisingly, Domhan et al. (2015)'s method takes up most of the time, as it requires significantly longer learning curves to make its decision. In addition, we can confirm the results of Baker et al. (2018); Chandrashekaran & Lane (2017), both of which report better results than Domhan et al. (2015). Our method requires the least amount of time for each data set. For CIFAR-100 we do not observe any regret, but a reduction of the time by a factor of 100. In some cases we observe an insignificantly higher regret than some of the other methods. In our opinion, the time saved makes up for it.

In Figure 4 we visualize the random search for SVHN. As you can see, many curves not only have similar behavior but also similar accuracy. For this reason, it is difficult to decide whether to discard a model safely, which explains the increased runtime. The only methods that do not show this behavior are Successive Halving and Hyperband. The simple reason is that the number of runs terminated and the time they are discarded are fixed by the algorithm and do not rely on the properties of the learning curve. The disadvantages are obvious: promising runs may be terminated and unpromising runs may run longer than needed.

Finally, we compare our method with DARTS (Liu et al., 2019), a method that accelerates the search by using the parameter sharing. We train the architecture discovered in the previous experiment with an equivalent training setup like DARTS for a fair comparison. This architecture reached a 2.99% classification error on CIFAR-10 after only 20 GPU hours were searched. By comparison, DARTS (1st order) requires 36 GPU hours for a similarly good architecture (3.00% error). DARTS (2nd order) can improve these results to 2.76%, but requires 96 GPU hours.

## 4.4 LCRankNet prediction analysis

We saw in the previous selection that LCRankNet does not perform perfectly. There are cases where its regret is greater than zero or where the search time is higher which indicates that early stopping

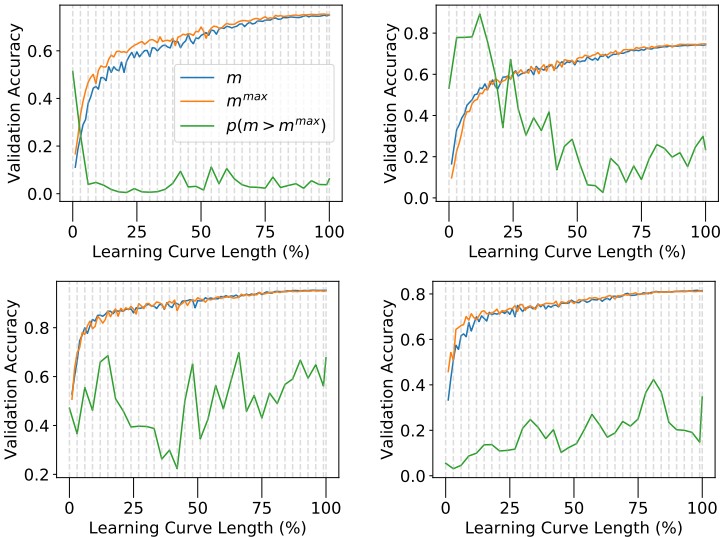

Figure 5: Analysis of the predicted probability of LCRankNet with growing learning curve length. The plots in the top row show examples for correct decisions by LCRankNet. The bottom row shows two examples where it's decision was wrong.

was applied later than maybe possible. In this section we try to shed some light on the decisions made by the model and try to give the reader some insight of the model's behavior.

We provide four example decisions of LCRankNet in Figure 5. The plots in the top row show cases where LCRankNet made a correct decision, the plots in the bottom row are examples for incorrect decisions.

In both of the correct cases (top row), LCRankNet assigns a higher probability to begin with, using meta-knowledge. However, in the top left case it becomes evident after only very few epochs that $m^{\mathrm{max}}$ is consistently better than $m$ such that the probability $p\left(m > m^{\mathrm{max}}\right)$ reduces sharply to values close to zero which would correctly stop training early on. The case in the top right plot is more tricky. The probability is increasing until a learning curve length of 12% as the learning curve seem to indicate that $m$ is better than $m^{\mathrm{max}}$. However, the learning curve of $m^{\mathrm{max}}$ approaches the values of $m$ and then the probability decreases and training is stopped early.

We continue now with the discussion of the two examples in the bottom row which LCRankNet false terminated early which cause a regret higher than zero in Table 1. In the bottom left we show an example where sometimes $m$ is better for a longer period and sometimes $m^{\mathrm{max}}$. This is arguable a very difficult problem and it is hard to predict which curve will eventually be better. However, LCRankNet shows a reasonable behavior. As the learning curve of $m$ is consistently worse than $m^{\mathrm{max}}$ in the segment from 15% to 42%, the probability is decreasing. Starting from a length of 57%, where $m$ shows superior performance than $m^{\mathrm{max}}$ for several epochs, the probability starts to raise. From learning curve length of 70% onwards, both the learning curves are very close and the difference in the final accuracy is only 0.0022. The example visualized in the bottom right plot is a very interesting one. The learning curve of $m^{\mathrm{max}}$ is consistently better than or almost equal to $m$ up to the very end. Towards the end, learning curves are getting very close. In fact, from learning curve length 63% onwards, the maximum difference between $m$ and $m^{\mathrm{max}}$ per epoch is only 0.008. Hence, in the beginning it seems like a trivial decision to reject $m$. However, eventually $m$ turns out to be better than $m^{\mathrm{max}}$.

In conclusion, deciding whether one model will be better than another one based on a partial learning curve is a challenging task. A model that turns out to be (slightly) better than another one can be dominated consistently for most of the learning curve. This makes it not only a very challenging problem for automated methods but for human experts as well.

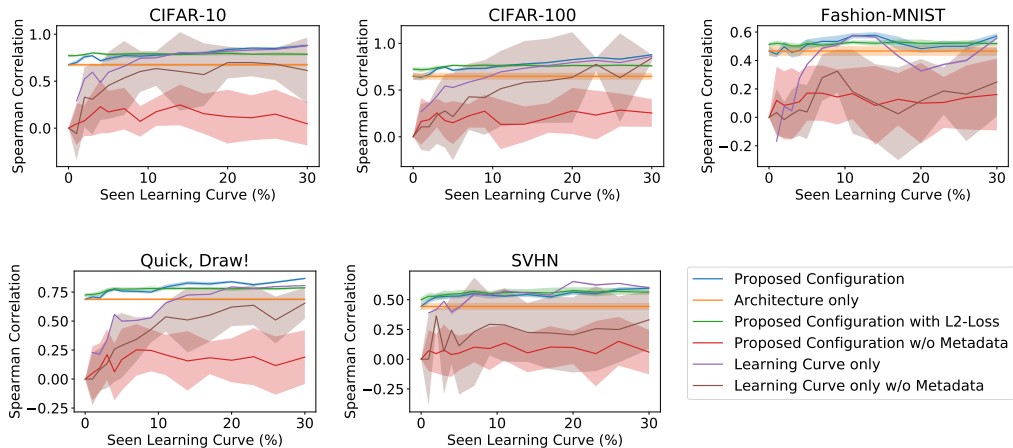

Figure 6: Analysis of the different components of LCRankNet. Every single component, metadata, consideration of the learning curve and architecture description, is vital.

## 4.5 ANALYSIS OF LCRANKNET'S COMPONENTS

We briefly mentioned before which components of our learning curve ranker have an essential influence on its quality. We would like to deepen the analysis at this point and compare the configuration we have proposed with different variants in Figure 6. We consider variants with and without metadata, architecture description or learning curve consideration. In addition, we compare our configuration trained with pairwise ranking loss to one trained with a pointwise ranking loss.

One of the most striking observations is that the metadata is essential for the model. This is not surprising since in particular the learning of architecture embedding needs sufficient data. Sufficient data is not available in this setup, so we observe a much higher variance for these variants. Even the variant that only considers the learning curve benefits from additional meta-knowledge. But even this is not surprising, since stagnating learning processes show similar behavior regardless of the data set. Using the meta-knowledge, both components achieve good results on their own. It can be clearly seen that these components are orthogonal to one another. The variant, which only considers the architecture, shows very good results for short learning curves. If only the learning curve is considered, the results are not very good at first, but improve significantly with the length of the learning curve. A combination of both methods ultimately leads to our method and further improves the results. Finally, we compare the use of a pointwise ranking loss (L2 loss) versus a pairwise ranking loss. Although our assumption was that the latter would have to be fundamentally better, since it optimizes the model parameters directly for the task at hand, in practice this does not necessarily seem to be the case. Especially for short learning curves, the simpler method achieves better results. However, once the learning curve is long enough, the pairwise ranking loss pays off.

## 5 CONCLUSION

In this paper we present LCRankNet, a method to automatically terminate unpromising model configurations early. The two main novelties of the underlying model are that it is able to consider learning curves from other data sets and that it uses a pairwise ranking loss. The former allows to predict for relatively short, and in extreme cases even without, learning curves. The latter directly allows to model the probability that one configuration is better than the another. We analyze our method on five different data sets against three alternatives. In an experiment to optimize network architectures, we obtain the fastest results. In the best case, LCRankNet is 100 times faster without sacrificing accuracy. We also examine the components and predictions of our method to give the reader a better understanding of the design choices and functionalities.

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
