# OpenReview forum: "Learning to Rank Learning Curves"
_ICLR.cc/2020/Conference — Reject_

### Official Review · AnonReviewer1 · 2019-10-22
**Official Blind Review #1**

**Rating:** 6

**Review:**


The paper proposes a new method to rank learning curves of neural networks which can be used to speed up neural architecture search.
Compared to previous work, the learning curve model not only takes hyperparameter configurations into account, but by training it on offline generated data, it is able to model learning curves across different datasets.
Applied to a simple neural architecture search strategy, the proposed method achieves higher speed-ups on image classification tasks than other methods from the literature.

While the method seems interesting, I don't think the paper is ready for acceptance yet, since it a) misses some important details and b) the empirical evaluation is not sufficient.
More precisely the following points need to be addressed:

 - In section 3, the paper says that all automated methods follow broadly the same principal that the first model is trained to completion. This is not correct, commonly used methods, such as Hyperband (Li et al.) or BOHB (Falkner et al.), use successive halving (Jamieson et al.) which trains a batch of configurations for just a minimum budget and then already discards poorly performing configurations. I also miss a discussion of these methods in the related work section.

Hyperband: A Novel Bandit-Based Approach to Hyperparameter Optimization
Li, L., Jamieson, K., DeSalvo, G., Rostamizadeh, A., and Talwalkar, A.
Journal of Machine Learning Research 2018

BOHB: Robust and efficient hyperparameter optimization at scale
S Falkner, A Klein, F Hutter
Proceedings of the 35th International Conference on Machine Learning

Non-stochastic best arm identification and hyperparameter optimization
K Jamieson, A Talwalkar
Artificial Intelligence and Statistics, 240-248


 - Is the learning curve model updated during the optimization process with the new observed data? If not, is there any way the model can adapt to new data? Also, what happens if learning curves are fundamentally different to the training data, e.g if different learning rate schedules are used than for generating the offline data?

 - A simple baseline that is missing, is to use the last observed value as approximation for the final performance which often works competitively to more powerful learning curve extrapolation methods (e.g see Klein et al.).

 - The method is only applied for a simple random search strategy, however, in practice, one would use more sophisticated methods, such as for example Bayesian optimization. The question is, how effective is the proposed method in this setting, since the distribution of learning curves might be dramatically different and biased towards more well-performing configurations with almost identical learning curves.

- I think the experiments would be more convincing, if the method shows strong performance when deploy in commonly used NAS/HPO methods, such as Hyperband or successive halving. This should be straight forward, since decisions which configuration is promoted to a higher budget can be made based on the model instead of just the last observed value.

- What is delta in the experiments? How sensitive is this hyperparameter in practice?


Minor comments:

- Related to this paper is the work by Gargiani et al. which also models learning curves across datasets on offline generated data
 Probabilistic Rollouts for Learning Curve Extrapolation Across Hyperparameter Settings
 M Gargiani, A Klein, S Falkner, F Hutter
 arXiv preprint arXiv:1910.04522

- Figure 5: visually it seems that all learning curves are almost identical, maybe it would be better if the plot could zoom in at least for the final stage of training.



after the rebuttal
-----------------------

I thank the authors for answering my  questions. In the rebuttal the authors addressed my concerned about the insufficient empirical evaluation and included other relevant baselines. I will increase my score.

**Experience Assessment:**

I have published in this field for several years.

**Review Assessment: Checking Correctness Of Derivations And Theory:**

N/A

**Review Assessment: Checking Correctness Of Experiments:**

I carefully checked the experiments.

**Review Assessment: Thoroughness In Paper Reading:**

I read the paper thoroughly.

---

> ### Author Response · Authors · 2019-11-08
> **Addressing your comments in the revised version**
>
> Thank you very much for your feedback.
>
> In the uploaded revised version we hope to have addressed all of your comments. We discuss the listed related work and conducted additional experiments to compare to Successive Halving and Hyperband. We further added the last observed value baseline.
>
> The model is retrained whenever new data is collected.
>
> In our experiment the training protocol is not part of the search. This is a common setup in the NAS community. In order to consider this as well, we require the learning rate to be another input for the model and have training data with different learning rates. Fundamentally different learning rates due to different learning rates will be a challenge for our method. However, there is no existing work we are aware of that would not fail completely if the learning rate schedule is suddenly changed. Lets assume we use a learning rate schedule with restarts and different runs can have restarts at different intervals. All methods based on Successive Halving will fail because it will terminate those runs which have been restarted most recently. Also Domhan et al. assume a monotonously increasing learning curve and would not be able to fit such learning curves. All other works would only work if sufficiently many learning curves with different settings are observed. But this is also the scenario where our method will work. We hope to follow up on this discussion if you do not agree with us.
>
> Bayesian optimization is able to manage the trade-off between exploring and exploiting the search space. A random search can be considered a search that only explores. In our experiment we sampled 200 architectures from more than 10^11 candidates at random. We believe that in this scenario even a more advanced search algorithm will be still in full exploration mode and not behave different to random search. However, the experiments for SVHN give you an idea what will happen if most learning curves are very similar which is probably the scenario you have in mind. As you see, more time is spend because it is harder to rule out specific learning curves early on. This is a general problem for all learning curve prediction methods. If really the case happens that BO is in full exploitation mode, one would probably consider to stop the search soon.
>
> We believe that we can freely decide how our algorithm operates. We will consider to create a version which runs in a Successive Halving setting in the future. We have no influence on how the baselines work. To have a fair comparison to our initial baselines, we followed the very same setup. Based on the reviewers requests, we added Successive Halving and Hyperband. However, they don't perform outstanding such that we see no evidence that our method would perform better in this setup.
>
> Delta allows to trade-off precision and recall. It allows to decrease the training time at the risk of increasing the regret. Therefore you cannot really consider it sensitive it is more some lever you can use to change the behavior of the algorithm. We added a more elaborate discussion to the paper.
>
> We are happy to answer any follow-up questions.

---

### Official Review · AnonReviewer2 · 2019-10-23
**Official Blind Review #2**

**Rating:** 3

**Review:**

Learning curve prediction methods try to predict the final performance of a candidate model before it gets fully trained. In this way, learning curve prediction can act as a fast method for performance measurements in AutoML. Compared with previous approaches, the proposed method allows for transferring useful knowledge among different datasets.

While the proposed method is simple, the authors have shown promising results. However, I have some concerns about the motivations and usage of the proposed method. Please see the questions below:

Q1. Learning curve prediction is a general approach that can be combined with many zero-order optimization methods. Does this paper focus on learning curve prediction on general AutoML problems or just NAS?
- If the authors want to target general AutoML problems, they should perform experiments with (1) more datasets (come from various domains) and consider more (2) search algorithms.
- For (1), the authors can follow Auto-sklearn and Auto-weka, or some experiments on graph (e.g., GCN) or some experiments on language modeling. For (2), they can combine the proposed approach with Bayesian optimization or genetic algorithms. CNN has nice transfer learning ability, to show the wide applicability of the proposed approach is important.
- If authors want to target at NAS, then the proposed method is not useful. Parameter sharing is a better method in NAS for fast performance evaluation. The authors need to compare with some recent NAS papers in CV (e.g., DARTS).

Q2. "Learning curve component". Learning curves are naturally a sequence of data points. The proposed method simply does a maximum pooling over all positions, which ignores the sequence of data. Could the authors give some explanations? Is it better to carry an ablation study on this point?

Q3. "Architecture component". How will the changes in the search space affect the proposed method?

Q4. Could the authors random draw many architectures from "NASNet search space" and then plot their learning curve? It is better to show what kinds of curves will the proposed method likely to give early stops.

Q5. Another main pitfall of the proposed method cannot offer a probabilistic estimation, which can be done by Baker et al. (2018). Since the model is early stopped, it is naturally that there are some uncertainty in the estimated ranks.

**Experience Assessment:**

I have published one or two papers in this area.

**Review Assessment: Checking Correctness Of Derivations And Theory:**

I assessed the sensibility of the derivations and theory.

**Review Assessment: Checking Correctness Of Experiments:**

I assessed the sensibility of the experiments.

**Review Assessment: Thoroughness In Paper Reading:**

I read the paper at least twice and used my best judgement in assessing the paper.

---

> ### Author Response · Authors · 2019-11-08
> **Addressing your questions and results for DARTS**
>
> We are very grateful that you took your time to review our work.
>
> Q1. We think that our method is applicable for both NAS and general ML but we admit that we only conducted experiments for NAS. Therefore, it is fair to assume that this paper focuses on NAS. In this short time we are not able to conduct an experiment for general ML. However, we are currently preparing a comparison against state-of-the-art NAS methods that leverage parameter sharing. We hope to finish this experiment over the weekend.
>
> Q2. Assuming that the learning curve is monotonous increasing just the maximum is exactly what all methods based on successive halving use. Furthermore, the last seen value is a strong baseline as pointed out by reviewer 3. For these two reasons alone this kind of modelling makes a lot of sense to us. However, we additionally added convolutional layers. They allow the model to learn to use first and second order gradients of the learning curve if this is considered useful. The resulting embedding vector is then used in another layer which may decide how important the features are. What ablation study would you have in mind? The problem with using all features is that the model will overfit. What we could do is use max pooling with strides. Since the architecture as modelled worked fine we did not consider changing it. As our ablation study shows, the learning curve component turns out to be important in its current form. We acknowledge that there might be scope for further improvement.
>
> Q3. A change in the search space will lead to a change in the representation. If this representation is again a sequence of tokens, our model will not change at all. If you refer to whether this model can transfer to new search spaces, then likely not. However, all learning curve prediction methods that are data-driven face this problem. That is a general property of machine learning algorithms. We refer at this point to the discussion with reviewer 3.
>
> Q4. We think this was addressed with figures 4 and 5 and their discussion. Can you outline in more detail what experiment you expect?
>
> Q5. We agree that there is always some uncertainty and that's why we investigated this issue and the probabilistic estimation of our model in section 4.4. We further compared our method to the work by Baker and empirically demonstrated to perform better. Can you outline in which case Baker would be able to outperform our method?
>
> We are looking forward to answer your follow-up questions. We will provide the results for the comparison against DARTS as soon as possible.

---

> > ### Author Response · Authors · 2019-11-11
> > **DARTS results**
> >
> > We trained the architecture discovered with random search + LCRankNet using an equivalent training scheme for comparison to the DARTS outcome. This network achieves a classification error of 2.99% after searching for 20 hours. The authors report two results for DARTS. The first order version needs 36 hours and results in 3.00% error and the second order version needs 96 hours and results in 2.76% error. This result indicates that our method requires less time. More importantly, we reveal that early termination of training jobs can be as efficient as the parameter sharing scheme and hence identifying a new direction for efficient NAS. We are not aware of any NAS work claiming efficiency (by e.g. parameter sharing) that considers early termination (e.g. by using Hyperband or other related work) as a baseline. We strongly believe that this is a discussion worth to be held.
> >
> > We added this result and a short paragraph to the related work.
> >
> > As requested earlier, would you mind to elaborate on Q4? We would like to plot what you are asking for but as explained earlier we do not fully understand what you are asking for.

---

### Official Review · AnonReviewer3 · 2019-10-26
**Official Blind Review #3**

**Rating:** 6

**Review:**

This paper considers the problem of automatic early termination in hyper-parameter search and neural architecture search.  The authors propose to form this problem as learning curve ranking and transfer learning. Unlike most previous approaches for learning curve prediction, which estimates the probability of whether the current model is better than the current best model or not by first extrapolating the learning curve and then invoke a heuristic measure, this paper proposes to predict the probability directly. The pairwise comparison probability is modeled as the logistic function of the difference of a scoring function f, where f is modeled as a neural network with the learning curve data as the input, and constructed with three components, including a learning curve component, an architecture component and a data set component. The neural network f is trained with some meta dataset, and then the early termination is then decided based on this pairwise comparison probability. The paper applied the proposed early termination approach to the neural architecture search of five image classification data sets, and evaluated the performance in terms of the Spearman correlation of the learning curve ranking and the regret and time for the architecture search. It also analyzes the learning curve prediction characteristics through a few concrete examples, followed by some ablation analysis.

The proposed approach is novel to my knowledge, and the numerical performance is also satisfying and convincing. However, there are a few issues that this paper should better improve upon.

Firstly, the methodology description at the beginning of section 3 is not clear enough. In particular, the motivation of choosing logistic function in (1) is not explained, and how the calculation of p_{i,j} from the final learning curves is done is also not elaborated. Some of the terminologies are also not clearly defined or specified. For example, what does "posteriors" mean (cf. the line before (2))? What is the meta-knowledge \mathcal{D}? Is it the one used to train the neural network of f? There is also a small typo in line 2 of Algorithm 1, where d should be D probably.

Secondly, although the numerical experiments are convincing within the framework of learning curves prediction, they are not sufficiently convincing when it comes to the scope of the neural architecture search (NAS) or hyper-parameter optimization (HPO). Comparisons with state-of-the-art NAS and HPO algorithms that do not use learning curves prediction (e.g., HyperBand, learning-by-learning algorithms using LBFGS, etc.) are not mentioned or compared with. The authors may either want to add comparisons with those algorithms, or provide some more applications of learning curve prediction to showcase the flexibility of the proposed approach.

**Experience Assessment:**

I have read many papers in this area.

**Review Assessment: Checking Correctness Of Derivations And Theory:**

I carefully checked the derivations and theory.

**Review Assessment: Checking Correctness Of Experiments:**

I assessed the sensibility of the experiments.

**Review Assessment: Thoroughness In Paper Reading:**

I read the paper thoroughly.

---

> ### Author Response · Authors · 2019-11-08
> **Clarification of the method and addition of further baselines**
>
> Thank you very much for your time to prepare this review.
>
> We rewrote section 3 to make things a little bit more clear. The choice of the logistic function is common practice in the learning-to-rank community. We clarified that in the text and provided a reference. We elaborated the calculation of p_{i,j} as well. Basically, whenever model i is better than model j, p_{i,j} = 1, if they are equally good the value is 0.5 and if model j is better the value is 0. Whether one model is better than the other is defined by the given learning curves. With posterior we refer to p_{i,j}. See also the provided reference. \mathcal{D} is the training data for f, that is correct. Since we actually did not use this notation anywhere, we removed it to avoid confusion. Thanks for spotting the typo.
>
> We added a discussion Successive Halving, Hyperband and BOHB as suggested by the reviewers. We further added Successive Halving and Hyperband as additional baselines. Could you point us to the papers on learning-by-learning algorithms using LBFGS? We are happy to discuss this work as well. At the moment we are preparing a comparison to alternative NAS methods. This is still work in progress and we'll hopefully finish this over the weekend.
>
> Please let us know if you have further comments or ideas.

---

> > ### Comment · AnonReviewer3 · 2019-11-14
> > **Thanks for the modifications**
> >
> > I have read the updated draft and I think the modifications are great. But as my concerns are mostly about improvement (but not correctness or contribution), I'd like to maintain my current rating for weak accept. For the learning-by-learning algorithms using LBFGS paper, I'm actually just generally referring to the series of papers on gradient-based hyper-parameter optimization [1,2] and learning-to-learn approaches for optimization algorithm learning [3]. Thanks!
> >
> > [1] Maclaurin, Dougal, David Duvenaud, and Ryan Adams. "Gradient-based hyperparameter optimization through reversible learning." International Conference on Machine Learning. 2015.
> > [2] Pedregosa, Fabian. "Hyperparameter optimization with approximate gradient." arXiv preprint arXiv:1602.02355 (2016).
> > [3] Andrychowicz, Marcin, et al. "Learning to learn by gradient descent by gradient descent." Advances in neural information processing systems. 2016.

---

### Decision · Program_Chairs · 2019-12-19

**Decision:**

Reject

**Comment:**

Authors propose a new way of early stopping for neural architecture search. In contrast to making keep or kill decisions based on extrapolating the learning curves then making decisions between alternatives, this work learns a model on pairwise comparisons between learning curves directly. Reviewers were concerned with over-claiming of novelty since the original version of this paper overlooked significant hyperparameter tuning works. In a revision, additional experiments were performed using some of the suggested methods but reviewers remained skeptical that the empirical experiments provided enough justification that this work was ready for prime time.